# How Do Telomere Abnormalities Regulate the Biology of Neuroblastoma?

**DOI:** 10.3390/biom11081112

**Published:** 2021-07-28

**Authors:** Jesmin Akter, Takehiko Kamijo

**Affiliations:** 1Saitama Cancer Center, Research Institute for Clinical Oncology, Saitama 362-0806, Japan; j.akter@saitama-pho.jp; 2Laboratory of Tumor Molecular Biology, Department of Graduate School of Science and Engineering, Saitama University, Saitama 362-0806, Japan

**Keywords:** telomere maintenance, TERT, ATRX, telomerase, ALT, neuroblastoma

## Abstract

Telomere maintenance plays important roles in genome stability and cell proliferation. Tumor cells acquire replicative immortality by activating a telomere-maintenance mechanism (TMM), either telomerase, a reverse transcriptase, or the alternative lengthening of telomeres (ALT) mechanism. Recent advances in the genetic and molecular characterization of TMM revealed that telomerase activation and ALT define distinct neuroblastoma (NB) subgroups with adverse outcomes, and represent promising therapeutic targets in high-risk neuroblastoma (HRNB), an aggressive childhood solid tumor that accounts for 15% of all pediatric-cancer deaths. Patients with HRNB frequently present with widely metastatic disease, with tumors harboring recurrent genetic aberrations (*MYCN* amplification, *TERT* rearrangements, and *ATRX* mutations), which are mutually exclusive and capable of promoting TMM. This review provides recent insights into our understanding of TMM in NB tumors, and highlights emerging therapeutic strategies as potential treatments for telomerase- and ALT-positive tumors.

## 1. Introduction

Neuroblastoma (NB) is the most common extracranial malignant solid tumor in infancy and is derived from cells of the embryonal neuronal crest [1,2]. It is a very heterogeneous disease in terms of outcomes and response treatments, from spontaneous regression (~50% of infants) to widely disseminated tumors that are frequently resistant to multimodal treatments [3,4,5]. Despite multimodal therapeutic treatments, the majority of relapsed high-risk neuroblastoma (HRNB) patients still succumb to the disease. The genomic amplification of *MYCN* was strongly associated with unfavorable patient outcomes in approximately 20% of primary NB tumors and 40% of HRNB. However, since the amplification of *MYCN* only occurs in 40% of HRNB, other genetic and/or epigenetic alterations may play an important role in the remaining 60% of this disease subtype. In addition to *MYCN* amplification, other segmental chromosomal aberrations, including 1p deletion, 11q deletion, and 17q gain, have been detected in HRNB. The gain of chromosome 17q and the loss of chromosome 1p were observed in one-half and one-third of NB cases, respectively, and are associated with an adverse patient outcome [4,5,6,7]. The loss of 11q is also observed in about one-third of NB tumors and is a marker of poor prognosis. In recent years, several massive parallel sequencing studies identified *MYCN*, *TERT*, and α-thalassemia/mental retardation syndrome X-linked (*ATRX*) aberrations as frequent and mutually exclusive drivers in HRNB [8,9,10]. Due to these three frequent genomic alterations, which converge to activate telomere maintenance mechanisms (TMM) [8,9,11], telomeres and TMM have emerged as pivotal attributes and indicators of poor prognosis for HRNB, with longer telomere length being associated with a worse prognosis [12,13,14]. In contrast, the absence of any detectable TMM was associated with spontaneous regression and excellent survival [13], supporting the beneficial effects of targeting TMM pathways for patients [8,9,15].

Clinical therapeutic strategies in NB are established on the basis of risk classification. Patients with low-risk disease spontaneously regress or undergo surgical resection, but chemotherapy is performed when a residual tumor is found or when surgical removal is difficult. For the intermediate-risk group, treatment options include chemotherapy for the progressive disease after surgery and/or relapse, an emergency irradiation for the cases with neurological symptoms. HRNB, on the other hand, is very difficult to treat and requires multimodal therapy to achieve the current survival rate of slightly less than 50%. HRNB is currently treated with a number of DNA-damaging agents, anti-GD2 antibodies, immunity checkpoint inhibitors targeting PD-L1, 13-cis-RA, ALK inhibitors, and CAR-T therapy [5]. Despite significant advances in the field of NB therapy, HRNB continues to have poor prognosis. There are no available clinical agents to effectively target TMM in NB. We briefly review recent knowledge on TMM and discuss the potential therapeutic purposes of targeting these regulatory mechanisms in NB for the future development of targeted agents for TMM.

## 2. TMM: A Distinct Prognostic Factor for High-Risk NB

Telomeres are specialized DNA-protein structures that cap the end of each linear chromosome to promote genomic stability while playing a key role in controlling cellular proliferation [16]. Human telomeric DNA comprises a variable short tandem repeat, mainly 5′-TTAGGG-3′, and has a double-stranded portion (range: 5–15 kb) that terminates in a single-stranded 3′-G-rich overhang (G-tail) of 150–200 nucleotides [17,18]. In each cycle of DNA replication, DNA polymerase alone cannot fully replicate linear telomere ends. Therefore, telomere ends progressively shorten after each cell division. These loss of telomere repeats are cumulative, leading to eventual chromosomal instability and senescence or apoptosis [19,20]. Tumor cells adopt TMM to prevent telomere shortening, acquire replicative immortality, and represent a malignant hallmark of several cancer cells [21]. In NB, telomeres are frequently maintained by telomerase activation, which results in amplified *MYCN* (*MYCN* transcriptionally activates the *TERT* gene), *TERT* rearrangements, or somatic mutations in the *TERT* promoter [9,22]. However, 20–25% of NB tumors utilize the telomerase-independent alternative lengthening of telomeres (ALT) to replenish telomere DNA, which depends on homologous recombination [23,24].

Previous studies reported a close relationship between telomere lengths and the prognosis of NB, with a longer telomere length being associated with a worse prognosis (*p* = 0.007) [14]; the overall-survival (OS) rate for cases of NB with telomere lengthening was 27%, which was significantly lower than that for cases with telomere shortening (89%) (*p* = 0.013) [25]. Moreover, the intratumoral diversity of telomere lengths in individual NB tumors was strongly associated with disease progression and death, and represents a novel biomarker of a poor prognosis [26]. A recent mechanistic classification of 208 primary NB, derived from the extensive profiling of a large set of NB specimens, pointed to TMM as a key prognostic indicator [13]. In addition, this group showed the unfavorable prognostic impact of TMM in combination with RAS and/or *TP53* pathway mutations in NB, and correlation between high expression levels of the *TERT* gene and TMM (Figure 1) [13]. On the basis of TMM, Koneru et al. [15] divided HRNB patients into three subgroups (*TERT*-high, ALT-positive, and *TERT*-low/non-ALT). Consistent with previous findings [23], this group identified ALT in 23.4% of patients with HRNB, who had poor clinical outcomes. Five-year OS rates in patients with *TERT*-high and ALT tumors were 28 and 46%, respectively, while TMM-negative patients (*TERT*-low/non-ALT) had a significantly higher long-term OS rate of 75% [15]. However, >10% [23] or 12%–26% [15] of HRNB tumors had the ever-shorter telomere (EST) phenotype with a high telomere content and continually shortening telomeres due to the lack of TMM, low *TERT* expression levels, and no ALT, and these patients had significantly better OS. The mechanisms by which these tumors survive without activating one of the two known TMM remain unclear. Roderwieser et al. [27] recently reported that telomerase activation and ALT define distinct NB subgroups with adverse outcomes and that telomerase represents a promising therapeutic target in the majority of HRNB. In contrast, low-risk NB lacked genomic alterations in *TERT*, *MYCN*, or *ATRX*, suggesting the lack of TMM in these favorable tumors [9]. Although previous TMM studies on NB suggested that ALT-positive and TEL + NB both correlated with a high-risk tumor status and poor clinical outcomes, recent studies confirmed the presence of telomerase to be less favorable than that of ALT [15,26,27]. Collectively, these findings provide support for correlation between TMM and the underlying molecular pathogenesis of NB, and are of biological and clinical relevance (Table 1 and Table 2) [13,15,23,27,28].

## 3. Telomere Maintenance by Telomerase Activation

Telomerase is a functional ribonucleoprotein enzyme complex that is responsible for maintaining telomere lengths by synthesizing telomeric DNA repeats at the 3′ ends of linear chromosomes, thereby compensating for telomere attrition during each round of DNA replication [29]. It is a reverse transcriptase that consists of two essential components: a functional catalytic protein subunit called human telomerase reverse transcriptase (hTERT) encoded by the *TERT* gene, and an essential RNA component known as human telomerase RNA (hTERC or hTR), encoded by the *TERC* gene. The limiting factor for telomerase activity is *TERT* expression, which is detectable in more than 90% of human malignancies [30].

Normal human somatic cells exhibit very low or undetectable telomerase activity. In contrast, the overexpression of *TERT* was previously identified in 73% of all cancers by systemic analysis of *TERT* gene amplification in the Cancer Genome Atlas (TCGA) cohort including 6835 patients and covering 31 tumor types [31]. Tumor cells from several tumor types normally reactivate telomerase expression to maintain telomere lengths, thereby escaping senescence or apoptosis and promoting unlimited replication [30,32,33]. The up-regulation or reactivation of telomerase is a critical feature in the development of about 85% of human cancers [34], and is correlated with poor prognosis [14,25,35,36]. High telomerase expression levels define a large group of HRNB with increased invasiveness and poor prognosis [37], similar to *MYCN*-amplified tumors [38]. Telomerase may be activated by the induction of *TERT*, which may occur through at least two pathways: *MYCN* amplification and genomic rearrangements around *TERT*, including *TERT* promoter mutations, *TERT* structural variants (*TERT^SVs^*), and epigenetic modifications through *TERT* promoter methylation [8,9,14,27]. Immunofluorescence of 102 primary NB revealed the variable expression of *TERT* in 99 cases, 31 and 68 of which showed high and low *TERT* expression levels, respectively. High *TERT* expression levels were identified as a robust predictor of event-free-survival (EFS) and OS in NB patients, irrespective of age [26]. A recent study demonstrated that telomerase-activation-associated alterations occurred in approximately one-third of primary NB, were associated with poor patient survival, and were an independent prognostic marker in the multivariable analysis of *TERT* rearrangements in 457 pretreatment NB [27]. Approximately 10%–20% of untreated tumors harbored *TERT* rearrangements and the amplification of *MYCN*, respectively, both of which induced the expression of *TERT*, which is consistent with previous findings [9]. In addition, *TERT* mRNA levels were elevated in approximately 4% of untreated tumors that lacked genomic aberrations in *TERT* or *MYCN*. NB with *TERT* rearrangements, *MYCN* amplification, or high *TERT* expression without these alterations were associated with unfavorable prognostic variables and adverse patient outcomes [8,9,27].

Genomic studies on primary NB revealed the involvement of *TERT* rearrangements in telomere maintenance by the induction of telomerase activity. Using break-apart FISH on 457 pretreatment NB, *TERT* rearrangements were detected in 46 out of 457 tumors (10.1%) [27]. *TERT* rearrangements occurred exclusively in 18.1% of patients aged 18 months or older at diagnosis and were predominantly detected in 18.8% of Stage 4 cases and 11.4% of *MYCN*-nonamplified tumors with genomic losses at chromosomes 1p and 11q. In another study on 108 NB cases, *TERT* rearrangements were detected in 23% of stage 3 and stage 4 cases, regardless of the amplification of *MYCN* (37%) or *ATRX* mutations (11%), and were confirmed as an independent prognostic factor [8]. Peifer et al. [9] performed whole-genome sequencing on 56 primary NB and discovered recurrent genomic rearrangements in a 50 kb region upstream of the *TERT* transcriptional start site that did not affect the promoter on chromosome 5p15.33 in 12 of these cases. All tumors bearing *TERT* rearrangements harbored high *TERT* expression levels, which were achieved by juxtapositions of the *TERT*-coding sequence with super-enhancer elements [8,9]. *TERT* rearrangements only affected high-risk tumors and occurred in a mutually exclusive manner with the amplification of *MYCN* and deletion of *ATRX*. In an extended case series (*n* = 217), *TERT* rearrangements were detected in 28 cases (13%), of which 27 (24%) were high-risk and Stage 4 NB patients with a poor clinical outcome [9]. Telomeres were significantly longer in *TERT*-rearranged NB than those in nonrearranged NB. Therefore, *TERT* rearrangements are the second most frequent gene defect in HRNB after *MYCN* alterations and define a high-risk subgroup of NB.

*MYCN* amplification is a powerful prognostic indicator for HRNB that is associated with high telomerase activity and the upregulation of *TERT* [9,22]. Since MYC-binding sites have been detected in the *TERT* promoter region, the overexpression of MYCN appears to promote telomere stabilization via telomerase activation through a transcriptional increase in the expression of *TERT* [39,40,41]. In a large cohort of 379 NB tumors, the amplification of *MYCN* and *TERT* rearrangements was strongly associated with elevated *TERT* mRNA levels [13]. Moreover, Peifer et al. [9] reported the strong upregulation of *TERT* expression in *MYCN*-amplified tumors, whereas HRNB without *TERT* or *MYCN* alterations had low *TERT* mRNA levels and lacked telomerase activity and the activation of the ALT pathway.

*TERT* structural variations involving rearrangements of the *TERT* gene are associated with the strong induction of *TERT* expression, suggesting that these alterations regularly contribute to TMM and the immortalization of tumors. Similar to other studies, Koneru et al. [15] identified *TERT^SV^* in approximately 20% of HRNB patients, and all *TERT^SVs^* tumors had highly upregulated *TERT* mRNA. The activation of *TERT* in HRNB was not limited to tumors with *TERT^SVs^* or the amplification of *MYCN* because some tumors without these alterations expressed *TERT* as highly as *TERT^SVs^* and *MYCN*-amplified tumors [13,15]. In addition, *TERT* expression levels in some *MYCN*-amplified tumors were as low as those in ALT tumors. MYCN overexpression studies revealed that MYCN upregulated *TERT* expression via an intact and nonrepressed *TERT* promoter, but was not sufficient to overcome *TERT* repression in telomerase-negative and *TERT^SVs^*-positive cell lines, indicating an unidentified mechanism for the activation of *TERT* in NB. RNA-seq of *TERT* expression and the C-circle assay to detect ALT showed that 12%–26% of HRNB tumors (including some *MYCN*-amplified tumors) had low *TERT* expression levels and lacked ALT activation, and these patients had significantly better OS.

The incidence of hotspot mutations in the *TERT* promoter driving telomerase activity varied from undetectable to more than 90% in various human malignancies [42]. *TERT* promoter hotspot mutations were detected 124 bp (C228T) and 146 bp (C250T) upstream of the transcriptional start site ATG [43,44], and are the most common alterations related to the upregulation of telomerase [45]. In contrast to other cancers, mutations in the *TERT* promoter region are rare in NB primary tumors and cell lines [9,46]. *TERT* promoter mutations were not detected in a large series (*n* = 131) of primary NB; however, these mutations were only searched for in the core promoter region, which did not exclude their potential presence in more distant regulatory regions.

## 4. Telomere Maintenance by ALT

In the absence of telomerase activity, tumor cells maintain functional telomeres by utilizing an alternative route of TMM, namely, ALT. A lower proportion of human tumors (approximately 10%–15%) adopt ALT over telomerase reactivation to potentiate their replicative immortality [47,48,49,50]. Although ALT occurs in common tumors, it is the most prevalent in tumors of mesenchymal origin, including those arising from bone, soft tissue, neuroendocrine systems, and the peripheral and central nervous systems, and is generally associated with a poor prognosis [50,51,52]. Approximately 25%–30% of NB employ the ALT pathway, which is generally associated with unfavorable NB in older children without *MYCN* amplification and independent of telomerase activation status [8,9,15]. These tumors have a very poor clinical outcome. Another study that used telomere length as a marker of ALT indicated that this mechanism may occur in up to 59% of NB tumors [26].

The coexistence of ALT and telomerase was reported in various tumor types [53,54,55]. Some antitelomerase-based treatments demonstrated the ability of some tumor cells to escape death and switch from telomerase to ALT [53,56,57]. However, the mechanisms underlying possible switching or the co-existence of telomerase and ALT within the same cell or different heterogeneous cell subpopulations in a tumor remain unclear. ALT patient tumors, cell lines, and PDXs in NB expressed very low *TERT* mRNA levels, indicating that ALT and telomerase activation occur in a mutually exclusive manner [15] rather than coexist. In accordance with these findings, the recent profiling of ALT-positive NB tumors showed minimal to no *TERT* mRNA expression linked to low telomerase activity, resulting from epigenetic silencing of the *TERT* locus by H3K27me3 [28]. In contrast, Pezzolo et al. [26] suggested that the coexpression of the ALT mechanism and *TERT*, observed in 60% of analyzed NB tumors, plays a major role in NB tumor progression. Moreover, the coexistence of cancer-cell subpopulations with different telomere lengths within NB correlated with a poor clinical outcome and disease progression in NB patients.

ALT is a homology-directed recombination-dependent replication pathway that utilizes telomeric DNA as a template for elongation. Although the molecular mechanisms by which ALT telomere maintenance occurs remain unclear, previous findings indicated that ALT telomeres were prone to replication stress and that double-strand breaks caused by replication fork collapse may give rise to break-induced telomere synthesis, resulting in long tract telomere extensions of up to 70 kb [30,50,58,59,60]. A recent study suggested that ALT is a bifurcated pathway involving RAD52-dependent and RAD52-independent mechanisms mediated by break-induced DNA replication [61], very similar to that observed at double-strand breaks in yeast [62].

In addition to a lack of reliance on telomerase, ALT is characterized by a number of markers, including heterogeneous telomere length [63,64], a high level of telomere-sister chromatid exchange [65], a specialized telomeric nuclear structure called ALT-associated promyelocytic leukemia protein bodies (APBs) [66], and the presence of extrachromosomal telomeric DNA repeats in the form of partial double-stranded circles, termed C-circles [67,68]. Increased replicative stress and telomeric DNA damage-induced foci, a potential driver of the generation of ALT, are frequently observed in ALT-positive cells and regarded as a hallmark of ALT [69,70,71]. Recent studies identified two new possible markers of ALT: mitotic DNA synthesis [58,60] and the upregulation of long noncoding telomeric repeat-containing RNA (also called TERRA) [72]. A previous study [28] screened for ALT in primary and relapsed NB (*n* = 760), characterized its features using multiomics profiling, and suggested that ALT-positive tumors are clinically and molecularly distinct. ALT is clinically associated with a prolonged disease course and poor outcome. One of its molecular features is mutated *ATRX* and/or reduced protein abundance, heterochromatic telomeric chromatin, a slow proliferative capacity, and the frequent formation of neotelomeres.

## 5. ALT and *ATRX* Genetic Alterations

Although details on the ALT mechanism remain unclear, different factors may be involved in its activation during cell immortalization and cancer development. Loss-of-function (LoF) genetic alterations in the chromatin remodeling genes *ATRX* and *DAXX* (death domain-associated protein) were associated with ALT in multiple malignancies [30,73,74,75,76,77]. Moreover, ALT is less commonly associated with mutations in *TP53, IDH, H3.3 G34R/V*, *H3F3A*, and *SMARCL1* [78,79,80,81,82,83,84]. LoF mutations in *ATRX* are the most common genetic lesions in NB [24,85]. Although frequently screened for, *DAXX* mutations are rare in NB. Approximately 50% of ALT NB are associated with somatic alterations in *ATRX* [8,9,15,23,28]. ALT was significantly more frequent in *ATRX*-mutant NB than that in *ATRX* wild-type tumors [89.5% (17/19) vs. 22.2% (4/18), *p* < 0.0001] [10]. ALT was examined in a cohort of 149 NB [23], and detected in approximately 25% (36/49) of tumors on the basis of the C-circle assay. It was not present in *MYCN*-amplified tumors and correlated with poor outcomes in NB. The whole-exome and -genome sequencing of 13 ALT-positive tumors revealed that 8 had an *ATRX* mutation, 1 had a *DAXX* mutation, and 1 had a *TP53* mutation, whereas 4 were the wild types for *ATRX* and *DAXX*. ALT tumors also had significantly higher relative telomere content than that of ALT-negative tumors (*p* < 0.0001), irrespective of age because ALT NB patients were significantly older at diagnosis than those with ALT-negative NB (*p* < 0.0001). Roderwieser et al. [27] assessed APBs, a hallmark of ALT-positive tumors, in 273 out of 457 NB. ALT activation was detected in 49 out of 273 cases (17.9%) and associated with unfavorable prognostic variables and adverse outcomes. APBs were mainly observed in the tumors of patients aged 18 months or older at diagnosis (29.6%) and predominantly in stage 4 tumors (25.2%). In contrast, APBs occurred mutually exclusively with *MYCN* amplification and *TERT* rearrangements. In addition, genomic alterations in *ATRX* were identified in 8 out of 109 evaluable cases (7.3%), all of which were positive for APB, whereas no *ATRX* mutations were detected in 12 additional APB-positive cases, and the genes involved in ALT were unknown in the remaining cases. Koneru et al. recently performed the C-circle assay as an ALT biomarker; 25 of 107 HRNB tumors (23.4%) were positive for ALT and genomic alterations in *ATRX* were detected in 13 out of 23 ALT tumors (57%). The remaining ALT tumors lacked *ATRX* genomic alterations as well as other known ALT-associated genes (*DAXX*, *H3F3A*, or *SMARCAL1*) [15], Moreover, non-ALT tumors (C-circle-negative) lacked *ATRX* genomic alterations, which were confirmed to be the wild type for *ATRX* using Sanger sequencing. ALT activation without *ATRX* genomic alterations was also observed in NB cell lines and PDXs. The ALT cell line SK-N-FI [25,27] and two ALT PDXs (COG-N-589x and COG-N-620x) were wild-type *ATRX* that expressed the ATRX protein, indicating that ALT occurs in NB with and without *ATRX* genomic alterations.

In a cohort [28] of primary and relapsed NB, 9.2% (66/720) and 47.5% (19/40) of tumors, respectively, were classified as ALT-positive, indicating the strong enrichment of this molecular subgroup in relapsed cases irrespective of the INSS stage. The activation of ALT was mutually exclusive to *TERT* rearrangements and 55% of ALT-positive tumors had LoF mutations in *ATRX*, including single nucleotide variants, large deletions, and focal intragenic *ATRX* duplications. Mutations in the *ATRX* complex members *DAXX* and *H3F3A* were extremely rare in this cohort. In contrast, somatic mutations in *TP53* pathway genes (*TP53*, *CREBBP*, *ATM*, *ATR*, *CDKN2A*, and *MDM2*) were significantly enriched (*p* = 0.01) in ALT-positive tumors. ALT-positive tumors had the highest prevalence of *CDK4* amplifications, higher *CCND1* mRNA expression levels, copy number losses in POLD3 and ATM, mutations in Synaptic nuclear envelope protein 1, and deletions in receptor-type tyrosine-protein phosphatase delta. Canonical activating RAS pathway mutations (*HRAS*, *NRAS*, *KRAS*, *BRAF*, *RAF1*, *CDK4*, *CCND1*, and *NF1*) were significantly more frequent in relapsed ALT-positive NB (*p* = 0.0013), supporting the specific impact of *RAS* pathway mutations on relapsed ALT-positive tumors. Previous findings also revealed that patients with *TERT* or ALT activation and harboring alterations in the *RAS*/*TP53* cellular pathway were very high-risk cases that were prone to relapse and had a very poor clinical outcome [13]. In addition, integrating proteomic profiling identified reduced ATRX protein levels as a biomarker of ALT-positive NB that is independent of mRNA levels and *ATRX* mutations [28]. Decreased ATRX protein levels despite an unchanged abundance of mRNA may be attributed to the reduced translation and/or increased degradation of ATRX in ALT-positive tumors, which may result from the downregulation of the DAXX protein. Consistent with this finding, the knockdown of DAXX decreased ATRX protein levels in NB cells. Interestingly, DAXX protein levels were significantly reduced in ALT-positive *ATRX* wild-type tumors, while mRNA levels did not significantly differ and no recurrent mutation patterns in DAXX were observed, which is consistent with previous findings [13]. Reduced DAXX protein levels impair the assembly of the ATRX/DAXX complex, which then induces the degradation of orphan ATRX protein molecules. Hence, further studies are needed to elucidate the mechanisms underlying ATRX/DAXX complex reductions.

Age is a powerful indicator of the clinical outcome of NB [86]. A pan-NB analysis of 702 NB samples by Brady et al. [87] revealed that *MYCN* and *TERT* alterations were enriched in younger patients (median age of 2.3 and 3.8 years, respectively), while *ATRX* was more common in older children (median age of 5.6 years). This group also showed age-associated mutual exclusivity between *ATRX* and *TERT* and between *ATRX* and *MYCN*, indicating the susceptibility of different ages to specific oncogenic events.

The loss of ATRX functions was recently shown to not only be mutually exclusive to the amplification of *MYCN*, but also incompatible with the overexpression of the MYCN protein due to the detrimental accumulation of RS. Zeineldin et al. [10] recently demonstrated that the knockout of ATRX decreased colony formation in *MYCN*-amplified NB cell lines, while no changes were observed in *MYCN* wild-type NB cell lines. Correspondingly, the induction of *MYCN* expression in *ATRX*-mutant NB cells and U2OS cells (*ATRX* mutation) resulted in the significant loss of viability. Electron microscopy studies showed that a concomitant *ATRX* mutation and *MYCN* amplification resulted in mitochondrial disruption. Therefore, the synthetic lethality of *MYCN* amplification and *ATRX* mutation warrants further study as a novel therapeutic strategy for the treatment of NB patients.

## 6. Therapeutic Strategy Targeting TMM

Telomere maintenance is a powerful prognostic marker of HRNB thereby representing an attractive target for the development of novel therapeutic treatments.

### 6.1. Therapies against Telomerase Activity

Preclinical studies established telomerase targeting as a novel therapeutic approach for telomerase-positive HRNB, suggesting that telomerase-interacting compounds need to be evaluated in clinical trials; however, no clinical candidates are available. Although there are currently no telomerase targeted therapies as a standard cancer treatment, telomerase remains a potential target for the development of novel therapies.

**Imetelastat (GRN163L)** is a competitive inhibitor of telomerase enzymatic activity, and its preclinical efficacy was evaluated in pediatric trials [88,89,90]. However, the clinical development of this compound was halted due to unacceptable toxicity in a Phase I trial on 20 children with refractory or recurrent solid tumors, including a patient with NB [91], and in a Phase II study on children with central-nervous-system malignancies [92].

**6-Thio-2′-deoxyguanosine (6-thio-dG)** is a nucleoside analogue that may be incorporated into telomerase-mediated de novo synthesized telomeric DNA in telomerase active cells, resulting in telomere dysfunction and rapid cell death [93]. This novel drug has promising preclinical efficacy against multiple tumors, including NB [27]. In vivo studies showed that 6-thio-dG incorporation created DNA damage and induced cell death in cancer-cell lines. The treatment of telomerase-activated NB cells with 6-thio-dG impaired cell viability at significantly lower concentrations, suggesting its on-target specificity. Moreover, 6-thio-dG strongly impaired tumor growth in NB xenograft models with *TERT* activation, but exerted weaker effects on *MYCN*-amplified NB, likely due to the additional oncogenic pathways activated by *MYCN*. Additional in vivo studies are needed to evaluate the antitumor effect of 6-thio-dG in *MYCN*-amplified NB. However, clinical trials are awaited because 6-thio-dG is less toxic than traditional telomerase inhibitors are [94], and is the highest priority compound for further development for NB.

**The G-quadruplex stabilizer, telomestatin** inhibits telomerase activity [95]. Laboratory-based studies demonstrated that the treatment of telomerase-expressing NB cell lines with telomestatin resulted in dose-dependent cytotoxicity and apoptosis via telomere shortening [35]; however, it is not yet in clinical development.

A recent study reported the efficacy of combination therapy using the BET bromodomain inhibitor **JQ1 or AZD5153**, targeting the BET bromodomain protein BRD4, and a CDK inhibitor (**dinaciclib**) [96,97]. BRD4 and CDKs play critical regulatory roles in the expression and chromatin activation of *TERT* and multiple *TERT*-associated genes. The epigenetic targeting of BRD4 or CDKs with their respective inhibitors suppressed the expression of *TERT* and multiple *TERT*-associated genes in NB with TERT overexpression and *MYCN* amplification. These two inhibitors in combination act synergistically or additively to inhibit the growth of NB cells in vitro and NB xenograft growth in vivo.

Another recent study examined the efficacy of the BET bromodomain inhibitor **OTX015** and **carfilzomib** (a proteasome inhibitor), which is an approved oncology drug [98]. OTX015 and carfilzomib exerted strong synergistic effects to block the overexpression of TERT, induced *TERT*-rearranged NB cell apoptosis in vitro, and strongly suppressed tumor progression in mouse models. These findings encourage the initiation of the first clinical trial based on combination therapy for patients harboring *TERT*-rearranged NB tumors.

### 6.2. Therapies against ALT

While extensive efforts are focused on telomerase targeting approaches, therapeutic strategies that effectively and specifically target ALT cancers are currently limited. A better understanding of the mechanisms and prevalence of ALT is needed to develop treatments and select eligible patients for targeted therapies. However, some studies reported potential therapeutic strategies for ALT.

**ATR inhibitor:** In osteosarcoma models, the presence of ALT rendered cells hypersensitive to ATR inhibition [99]. However, another study directly refuted these findings and concluded that ALT is not an independent factor influencing ATR inhibitor sensitivity [100].

George et al. [101] recently evaluated the clinical ATR inhibitor AZD6738 in a panel of telomerase- and ALT-positive NB and non-NB cell lines, and found that ALT-positive cells were generally not more sensitive to ATR inhibition than telomerase-positive cells are following ATR inhibition. These findings provide support for differences in ATR inhibitor sensitivity not being related to ATRX-deficient ALT [100].

**ATM inhibitor combination therapy:** ATM was recently shown to be hyperactivated at ALT telomeres and conferred chemoresistance to DNA-damaging agents (e.g., topoisomerase inhibitors) in ALT NB. The ATM inhibitor AZD0156, which is currently in adult early-phase clinical trials, exerted synergistic effects with temozolomide and irinotecan therapy in pre-clinical models of ALT NB [102].

**PARP inhibitor and DNA-damaging agent combination therapy:** Genetic alterations in *ATRX* have been detected in approximately 50% of ALT NB cases and are suggested to suppress the ALT pathway [103], and, thus, represent another important potential indirect therapeutic target. In 2020, George et al. [104] revealed that the inactivation of *ATRX* increased DNA damage and homologous recombination repair (HRR) defects while impairing replication fork processivity. Consistent with these findings, high-throughput compound screening showed the preclinical sensitivity of combination therapy with olaparib (a PARP inhibitor)/irinotecan (a DNA-damaging agent) in NB models with genetic alterations in *ATRX*, which may be rapidly translated into clinical settings [104].

**Tetra-Pt (bpy)**, a novel cisplatin derivative that targets the telomeric G-quadruplex, selectively inhibited the growth of ALT-cell xenografts tumors in mice [105]. However, this compound is not currently available for clinical trials and has not yet been evaluated in NB models.

## 7. Conclusions

TMM are a main target for the development of anticancer strategies due to their crucial role in cancer development (Figure 1). Although promising preclinical data regarding the therapeutic targeting of TMM have recently been obtained, there are currently no clinical trials available for these large molecular subgroups of high-risk patients. Consequently, anticancer therapies targeted to TMM must take into consideration whether one or more TMM are present in a tumor. Another concern with TMM targeted therapies is that cells may change their active mechanism when one becomes nonfunctional. Therefore, the targeting of telomerase as an anticancer therapeutic may select ALT-positive cells that may be resistant to these therapies, which supports the need to develop anticancer therapeutics against both TMM. Moreover, according to Dagg et al. [21], high malignancy may also develop in the absence of activated TMM. Therefore, tumors with the EST phenotype are resistant to drugs against TMM, and the existence of EST tumors underscores the need to assay for TMM prior to treatments with these drugs.

## Figures and Tables

**Figure 1 biomolecules-11-01112-f001:**
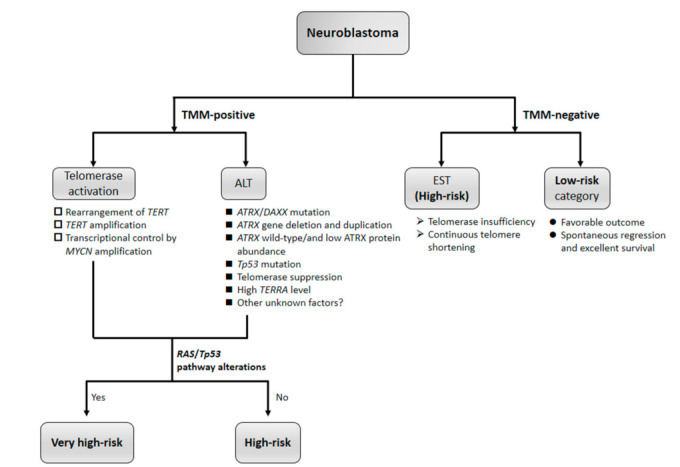
Telomere maintenance mechanisms (TMM) in NB. Presence and absence of TMM and *RAS* or *TP53* pathway alterations define clinical subgroups of NB.

**Table 1 biomolecules-11-01112-t001:** Incidence of TMM triggered by telomerase and ALT or no TMM in NB from several cohorts of different studies.

Study Group	Telomerase-Dependent TMM (%)	ALT-Dependent TMM (%)	TMM-Negative
Dagg et al. (2017) [23]		24% (36/149) C-circle-positive Non-*MYCN-*amplified (0/36) 49% (55/113) of ALT-negative tumors had the amplification of *MYCN* (*p* < 0.0001) *ATRX* mutation (8/13), *DAXX* mutation (1/13), *TP53* mutation (1/13) Mutual exclusivity between *MYCN* amplification and aberrant *ATRX* Higher relative telomere content Older age at diagnosis (median age 4.4 years)	Tumors and cells 29% (17/58) of high-risk tumors *MYCN* nonamplified/ALT-negative/telomerase-negative High telomere DNA content Initially had long telomeres Borderline C-circle levels Comprises >10% of high-risk patient tumors Older age at diagnosis (median age 4.67 years) These tumor groups are biologically distinct from Stage 4s group 51% of patients with these NB died of the disease despite therapy Termed ever-shorter telomere (EST) phenotype EST NB cell lines (LA-N-6 and COG-N-291) EST cells have very long telomeres and extensive proliferative capabilities Had wild-type *ATRX*, *DAXX*, *H3F3A*, or TP53 Continuous shortening of telomeres in EST cell lines Activation of telomerase or ALT may rescue the EST phenotype
Ackermann et al. (2018) [13]	*TERT^SV+^* (21/208) *MYCN* amplification (52/208) High *TERT* expression level (telomerase activation) Very high-risk with *RAS*/*TP53* pathway alterations Age ≥ 18 months, Stage 4	APB-positive (31/208) Low *TERT* expression level, high telomere length ratio Inactivated *ATRX* mutation No mutation in *DAXX* Very high-risk with *RAS*/*TP53* pathway alterations Age ≥ 18 months, Stage 4	27/41 low-risk NB tumors and the lack of *MYCN*/*TERT*/*ATRX* alterations Age < 18 months
Roderwieser et al. (2019) [27]	10.1% (46/457) *TERT^SV+^* High-risk tumors Age 18 months or older 18.1% (46/254) Elevated *TERT* mRNA level	17.9% (49/273) APB-positive Age 18 months or older at diagnosis 29.6% (47/159) Stage 4 tumors 25.2% (32/127) Lower-stage tumors 11.6% (17/146) Mutually exclusive with *MYCN* amplification (0/59) and *TERT* rearrangements (0/29) *ATRX* genomic alterations 7.3% (8/109) Low TERT expression levels and increased telomere length ratio	
Koneru et al. (2020) [15]	20.3% (12/59) *TERT^SV+^* 0% *TERTp* mutation High-risk tumors Age > 18 months Highest *TERT* mRNA expression level	23.4% (25/107) C-circle-positive High-risk tumors Non-*MYCN* amplified Age > 18 months Heterogeneous telomere length *ATRX* genomic alterations 57% (13/23)	Tumors, cells, and PDXs Approximately 12–26% of high-risk NB tumors TERT low/non-ALT (C-circle-negative, low frequency of APBs) Lacked telomerase activity Higher relative telomere content High telomere length No genomic alterations in ATRX, DAXX, H3F3A, or SMARCAL1 EST cell lines (LA-N-6, COG-N-291, CHLA-132, COG-N-509hnb, and COG-N-562hnb) Continuous shortening of telomeres in EST cell lines
Hartlieb et al. (2021) [28]		9.2% (66/720), screening cohort 47.5% (19/40), relapse cohort C-circle-positive Higher telomere content Minimal to no TERT mRNA expression (low telomerase activity) Higher TERRA (telomeric long noncoding RNA) level Mutually exclusive to TERT rearrangements Loss-of-function mutations in ATRX (55%) Low ATRX protein abundance (independent of ATRX mRNA levels and the mutation status)	

**Table 2 biomolecules-11-01112-t002:** Data reported by different studies evaluating TMM impact on clinical outcomes of NB.

Marker Analyzed	Clinical Endpoint	Outcome	Ref.
TMM (defined by *TERT* expression and ALT based on APB detection)	Event-free survival (EFS); disease-specific survival (DSS)	TMM-negative nonhigh-risk patients had a more favorable outcome than that of the TMM-positive group (5-year EFS, *p* < 0.001; 5-year DSS, *p* < 0.001). *RAS*/*TP53* pathway alterations did not affect clinical outcomes in TMM-negative patients (EFS, *p* = 0.702; DSS, *p* = not applicable). In contrast, co-occurrence of *RAS*/*TP53* pathway alterations with TMM was associated with more adverse 5-year EFS or DSS (*p* = 0.006; *p* < 0.001). In multivariate analysis, TMM and *RAS* or *TP53* pathway mutations were independent prognostic markers of NB (TMM, *p* <0.001; *RAS* and/or *TP53* pathway mutation, *p* = 0.001).	[13]
*TERT* expression (RNA-seq); ALT (C-circle assay and/or telomere content measurements by qPCR and a TRF analysis)	EFS; overall survival (OS)	*TERT*-high, ALT, and *TERT*-low/non-ALT (TMM-negative), each showed distinct OS (*p* < 0.001) by the Log-rank test, but not EFS (*p* = 0.137). The 5-year OS rate for *TERT*-high was 28% versus 46% for ALT and 75% for the TMM-negative group. TMM-negative patients had a significantly higher OS rate than that of patients in the *TERT*-high or ALT-positive groups (*p* < 0.001). In the multivariate analysis, *TERT*-high expression or ALT-positivity predicted poor OS (*p* < 0.001), independent of other known risk factors for NB.	[15]
ALT (defined by telomere content and C-circle measurement or a TRF analysis)	EFS; OS	Outcome of ALT-positive NB patients was as poor as that for *MYCN*-amplified NB (5-year EFS 28% versus 24%; 5-year OS 36% versus 28%). ALT-negative/long telomere NB had a poor survival rate that was not significantly different from that of ALT-positive tumors (5-year OS 49% versus 36%; *p* = 0.1908).	[23]
*TERT* rearrangement (Break-apart FISH); *TERT* expression (Microarray); ALT (APB occurrence)	EFS; OS	Outcomes of NB patients with *TERT* rearrangements were poor for both EFS (5-year; *p* < 0.001) and OS (5-year; *p* < 0.001), and similar to those of the *MYCN*-amplified group (5-year EFS, *p* = 0.552; 5-year OS, *p* = 0.830). The prognosis of ALT-positive patients was also as poor as that of patients with *MYCN* or *TERT* alterations in terms of EFS (*p* < 0.001), whereas OS was significantly better (*p* = 0.002). *TERT* rearrangements identified as an independent prognostic indicator of adverse OS (*p* = 0.050).	[27]
ALT (C-circle assay)	EFS; OS	ALT-positive patients had similar EFS (*p* = 0.64) to those with *MYCN*-amplified tumors, but significantly longer OS (*p* = 0.0016). There were no significant differences in OS (*p* = 0.076) or EFS (*p* = 0.9) between patients with ALT-positive tumors divided into high- and low-/intermediate-risk groups. No significant differences were noted in survival between *ATRX*-mutated and *ATRX* wild-type ALT-positive patients (EFS, *p* = 0.39; OS, *p* = 0.081).	[28]

## Data Availability

Not applicable.

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
