# Peer review of "How Do Telomere Abnormalities Regulate the Biology of Neuroblastoma?"

_biomolecules, 2021, doi:10.3390/biom11081112_

Round 1

Reviewer 1 Report

The authors provide a detailed review on telomere maintenance mechanisms in high-risk neuroblastoma and include resulting therapeutic options.

This is a very well written, comprehensive, and highly relevant review addressing fundamental pathogenic processes and treatment approaches in a serious and still often fatal malignancy of early childhood. Therefore, this summary could contribute to establishing novel targeted therapies essential to improve treatment, outcome and long-term survival of this condition.   

Only few aspects need to be added and improved, respectively, before the manuscript can be published in Biomolecules.

Introduction (lines 23-44):

1.

This section should integrate a short clinically oriented paragraph on the recent therapeutic advances in addition to chemotherapy, surgery and irradiation, i. e. the implementation of GD2-antibodies and the inhibition of ALK as the first targeted agents regularly used in HRNB.

2.

To cover also other lesions recurrently detected in neuroblastoma, chromosomal aberrations, i. e. loss of 1p, as genetic characteristics and potential expression of genetic instability in HRNB should be addressed.

Tables 1 & 2:

While the content of Table 1 and Table 2 is highly informative, the format should be revised.

3.

In Table 1, the presentation of the individual points in column 1 (Telomerase-dependent TMM (%)), 2 (ALT-dependent TMM (%)) and 3 (TMM-negative) are in part displaced and therefore confusing. This should be improved in the final version.

4.

Table 2 contains a lot of text. Especially the text in column 3 describing the outcome should be reduced and depicted in a more systematic manner.

Therapeutic strategy targeting TMM (lines 323-414):

5.

Considering the heading of this section including the word “strategy”, this part of the review appears rather descriptive and abstract. Of course, it is always particularly complex to design and perform clinical trials including children. However, it would be encouraging and more useful to strategically discuss the agents available and provide perspectives how these could be implemented in clinical trials and treatment strategies.   

Author Response

We would like to express our sincere gratitude to both reviewers for their insightful comments. In addressing their concerns, we feel that our revised manuscript is significantly strengthened. Please find a point-by-point response to reviewer comments below (reviewer comments italicized, responses in bold).

Introduction (lines 23-44):

  1. This section should integrate a short clinically oriented paragraph on the recent therapeutic advances in addition to chemotherapy, surgery and irradiation, i. e. the implementation of GD2-antibodies and the inhibition of ALK as the first targeted agents regularly used in HRNB.

Answer:

We greatly appreciate these suggestions, and have integrate a short clinically oriented paragraph in introduction part (Page 2, line 47-60).

  1. To cover also other lesions recurrently detected in neuroblastoma, chromosomal aberrations, i. e. loss of 1p, as genetic characteristics and potential expression of genetic instability in HRNB should be addressed.

 Answer:

We appreciate this comment and have now included other lesions recurrently detected in neuroblastoma in introduction part (Page 1, line 33-38).

Tables 1 & 2: While the content of Table 1 and Table 2 is highly informative, the format should be revised.

  1. In Table 1, the presentation of the individual points in column 1 (Telomerase-dependent TMM (%)), 2 (ALT-dependent TMM (%)) and 3 (TMM-negative) are in part displaced and therefore confusing. This should be improved in the final version.

Answer:

We have revised the Table 1 to make this clearer (Page 3-4).

  1. Table 2 contains a lot of text. Especially the text in column 3 describing the outcome should be reduced and depicted in a more systematic manner.

 Answer:

 Thank you for this suggestion. We have revised the Table 2 in new version (Page 4-5).

Therapeutic strategy targeting TMM (lines 323-414):

  1. Considering the heading of this section including the word “strategy”, this part of the review appears rather descriptive and abstract. Of course, it is always particularly complex to design and perform clinical trials including children. However, it would be encouraging and more useful to strategically discuss the agents available and provide perspectives how these could be implemented in clinical trials and treatment strategies.   

Answer:

We thank the reviewer for this insightful suggestion. Although, there are no clinical agents available to effectively target TMM in NB, some preclinical studies have shown effectiveness of targeted therapy against TMM. Here we summarized the current preclinical research focused on targeting TMM, which may be rapidly translated in to clinical settings.  

For therapies against telomerase activity, 6-Thio-2'-deoxyguanosine (6-thio-dG) is the excellent candidate in NB because of strong growth impairment in NB xenograft models with TERT activation. Clinical trials are awaited because 6-thio-dG is considered to be less toxic than traditional telomerase inhibitors.

However, more study is required for potential therapeutic strategies against ALT.

We have included such kinds of information in our manuscript (Page 9-11), especially in the 6-thio-dG paragraph (page 10).

Reviewer 2 Report

Please see attached PDF.

Author Response

We would like to express our sincere gratitude to both reviewers for their insightful comments. In addressing their concerns, we feel that our revised manuscript is significantly strengthened. Please find a point-by-point response to reviewer comments below (reviewer comments italicized, responses in bold).

  1. Page 2, first paragraph, suggest adding a bridging sentence to complete the concept of telomere erosion-induced chromosome instability: "Therefore, telomere ends progressively shorten after each cell division. These loss of telomere repeats are cumulative, leading to the eventual chromosomal instability and senescence or apoptosis”

Answer:

We thank the reviewer for this suggestion. We now include this sentence in Page 2, line 70-71).

  1. Page 5, second paragraph. There is a lack of concordance in quoting the percentage of telomerase+ versus ALT+ tumor types (from 73% to 85% to 90%). This discrepancy is likely resulted from the direct quoting of various surveys, and could confuse the novice readers.

Answer:

Thanks for your suggestion. We made change in our revised manuscript (Page 5, line 125-133).

  1. Page 5, second paragraph, “NB with TERT rearrangements, MYCN amplification, or high TERT expression levels occurred mutually exclusively and were related to similarly poor patient outcomes”

This quoted statement cannot be true, given that TERT rearrangements and MYCN

amplification both lead to high TERT expression. Suggest rewrite.

Answer:

Thanks a lot for your suggestion. We have rewrite in our revised manuscript (Page 6, line 151-153).

  1. The concept of TERT rearrangements is well defined, but TERTSV less so. Are these structural variants overlap with TERT rearrangements? The authors should add a sentence to explain the relations/differences between the two genetic changes in the TERT gene.

Answer:

According to the article, TERT structural varients overlap with TERT rearrangements. TERT structural variants comprise amplification and rearrangement of the TERT locus.

We have added a sentence to define the relations between the two genetic changes (Page 6, line 185).  

  1. The concept of aggressive neuroblastoma subtypes with no detectable TMMs and exhibits short telomeres is highly relevant to the current topic. As TMM is essential for the continuous proliferation of immortal cells, a short discussion on the yet unknown mechanism(s) could be helpful. For example, could a similar mechanism as used by the Type I survivors of TLC-/- in yeast be employed in these cases?

Answer:

We thank the reviewer for this exciting comment. This is very interesting topics. Recently two papers [Ref. 15 and Ref. 23] discussed about this phenomena. “Aggressive neuroblastoma subtypes with no detectable TMMs”-underlying mechanism, how these tumors survive remains unclear. We have included some information about this topic in our manuscript (Page 2-3, line 95-99).

However, Dagg et al [Ref. 23], speculated that there may be tumor types that do not require an activated TMM and immortalization because the progenitor cells start with long telomeres, or the tumors are relatively simple genetically (removing the need for many rounds of cell division), or the tumor conditions support a high surviving fraction.

We think more whole genome sequencing data, RNA seq data or methylation profiling of primary neuroblastoma tumor is necessary for understanding the possible molecular mechanism of NB aggressiveness without TMM.

According to “Type I survivors of TLC-/- in yeast” (recombination mediated telomere elongation in the absence of telomerase)-this mechanism is very similar to BIR (Break-induced replication) by which a double strand break can be healed. In human cancers, ALT occurs through a RAD52-dependent and a RAD52-independent BIR pathway, which is very similar to Type 1 survivors in yeast. We think further study is necessary to understand this mechanism. We have included this information in our manuscript (Page 7, line 241-244) with reference.